# Could the Combination of Two Non-Psychotropic Cannabinoids Counteract Neuroinflammation? Effectiveness of Cannabidiol Associated with Cannabigerol

**DOI:** 10.3390/medicina55110747

**Published:** 2019-11-18

**Authors:** Santa Mammana, Eugenio Cavalli, Agnese Gugliandolo, Serena Silvestro, Federica Pollastro, Placido Bramanti, Emanuela Mazzon

**Affiliations:** 1IRCCS Centro Neurolesi “Bonino Pulejo”, 98124 Messina, Italy; santa.mammana@irccsme.it (S.M.); eugenio.cavalli@irccsme.it (E.C.); agnese.gugliandolo@irccsme.it (A.G.); serena.silvestro@irccsme.it (S.S.); placido.bramanti@irccsme.it (P.B.); 2Dipartimento di Scienze del Farmaco, Università del Piemonte Orientale, Largo Donegani 2, 28100 Novara, Italy; federica.pollastro@unipo.it

**Keywords:** NSC-34 cells, amyotrophic lateral sclerosis (ALS), neuroinflammation, lipopolysaccharide (LPS), cannabigerol, cannabidiol

## Abstract

*Background and Objectives*: Neuroinflammation is associated with many neurodegenerative diseases, including amyotrophic lateral sclerosis (ALS). In this study, we investigate the anti-inflammatory, anti-oxidant, and anti-apoptotic properties of two non-psychoactive phytocannabinoids, cannabigerol (CBG) and cannabidiol (CBD). *Materials and Methods:* The motoneuron-like cell line NSC-34 differentiated by serum deprivation and with the additional treatment of all-trans retinoic acid (RA) is a valid model to investigate molecular events linked to neurodegeneration in ALS. *Results:* Pre-treatment with CBG (at 2.5 and 5 µM doses) alone and in combination with CBD (at 2.5 and 5 µM doses) was able to reduce neuroinflammation induced by a culture medium of LPS-stimulated macrophages. In particular, the pre-treatment with CBD at a 5 µM dose decreased TNF-α levels and increased IL10 and IL-37 expression. CBG–CBD association at a 5 µM dose also reduced NF-kB nuclear factor activation with low degradation of the inhibitor of kappaB alpha (IkBα). CBG and CBD co-administered at a 5 µM dose decreased iNOS expression and increased Nrf2 levels. Furthermore, the pre-treatment with the association of two non-psychoactive cannabinoids downregulated Bax protein expression and upregulated Bcl-2 expression. Our data show the anti-inflammatory, anti-oxidant, and anti-apoptotic effects PPARγ-mediated. *Conclusions:* Our results provide preliminary support on the potential therapeutic application of a CBG–CBD combination for further preclinical studies.

## 1. Introduction

Neuroinflammation is a key pathological event associated with many neurodegenerative diseases, including amyotrophic lateral sclerosis (ALS). Inflammation triggered by microglia plays an important role in promoting neurodegeneration by inducing the expression of pro-inflammatory factors [1].

ALS is a progressive neurodegenerative disease caused by the selective death of alpha motor neurons. Many events, including oxidative stress, neuronal inflammation, mitochondrial dysfunction, RNA processing errors, and protein misfolding, are implicated in the pathogenesis of ALS [2,3].

NSC-34 is a murine hybridoma cell line produced by the hybridization of neuroblastoma cells with motor neuron-enriched spinal cord cells [4]. All-trans retinoic acid (RA), in combination with serum deprivation, promotes neuronal differentiation in NSC-34, characterized by morphological progression (such as the neurite outgrowth) and functional molecular changes that are motorneuron-specific [5]. It is a well-validated in vitro model to investigate the physiopathological mechanisms associated with neurodegeneration in ALS [6].

Lipopolysaccharides (LPSs) induces the expression of inflammatory mediators involved in the neurodegenerative process and is a valid model usually used to reproduce neuroinflammation associated with neurodegeneration [7].

In recent years, many in vitro and in vivo studies have demonstrated the therapeutic potential of cannabinoids for the treatment of neuroinflammatory disorders due to their antioxidant and anti-inflammatory properties [8].

*Cannabis sativa* is a plant of the Cannabaceae family widely utilized for its psychoactive and therapeutic effects [9]. To date, more than 120 cannabinoids have been isolated and characterized. The most investigated cannabinoids are the delta-9-tetrahydrocannabinol (Δ9-THC), cannabidiol (CBD), cannabigerol (CBG), cannabivarin (CBV), and cannabidivarin (CBDV). Δ9-THC is the predominant psychotropic component of the plant. In the last years, CBG and CBD are the two non-psychoactive cannabinoids more studied due to their anti-inflammatory and antioxidant properties [10,11]. The anti-inflammatory and antioxidant effects of CBD on movement disorders have also been proved in several preclinical and clinical studies. However, the beneficial effects in many of these studies have been observed when CBD is combined with Δ9-THC [12,13]. On the other hand, no study of the combination of no-psychoactive cannabinoids is available. CBG and CBD exert their activity at multiple molecular sites [8]. They may exert their actions by binding class A G-protein-coupled receptors (GPCRs); cannabinoid receptor type 1 (CB1), the most abundant neuromodulatory receptor in the brain; and cannabinoid receptor type 2 (CB2), localized in the immune system among other tissue districts [14]. Cannabinoids could also activate the transient receptor potential vanilloid channel-1 (TRPV1), and serotonin receptor 5-HT1A, or modulate G protein-coupled receptors (GPCRs) [15]. Their effects may also include interactions with transcription factors, such as the nuclear factor (erythroid-derived)-like 2 (Nrf-2), the nuclear factor kappa B (NF-κB) or nuclear receptors of the peroxisome proliferator-activated receptor (PPAR) family [16,17].

CBG is a partial agonist of CB1 and CB2 receptors. Additionally, it is an α2-adrenoceptor agonist and 5HT1A receptor antagonist. It can activate TRPV1, TRPV2, and transient receptor potential ankyrin 1 (TRPA1) and antagonize the transient receptor potential cation channel subfamily M (melastatin) member 8 (TRPM8) [18].

CBD has a low affinity for both cannabinoid CB1 and CB2 receptors and behaves as a non-competitive negative allosteric modulator of CB1 receptors [19].

CBG and CBD have demonstrated to exert neuroprotection by activation of the nuclear receptors of PPAR-γ by a mechanism cannabinoid (CB)-receptors independent [20,21].

Following the above-mentioned data, in this study, we wanted to evaluate the ability of two non-psychoactive cannabinoids, CBG administered alone and in association with CBD, to counteract neuroinflammation induced on NSC-34 differentiated motoneurons upon exposure to LPS-stimulated RAW 264.7 cell-conditioned medium.

## 2. Materials and Methods

### 2.1. Extraction and Isolation of CBG and CBD

*Cannabis sativa* was provided by greenhouse cultivation at CREA-CIN, Rovigo (Italy) in accordance with their legal status (authorization SP/106 23/05/2013 of the Ministry of Health, Rome, Italy). CBG and CBD were isolated and purified (with greater than 99% purity) according to a standardized protocol to avoid any trace of THC [22,23].

### 2.2. NSC-34 Cell Culture and Differentiation

NSC-34 motor neurons cells line were purchased from Cedarlane/CELLutions Biosystems Inc. (Burlington, ON, Canada) and maintained in DMEM-high glucose medium (Sigma-Aldrich; Merck KGaA, Darmstadt, Germany ) supplemented with 10% fetal bovine serum (FBS) (Sigma-Aldrich; Merck KGaA) and penicillin/streptomycin antibiotics. Cells were incubated at 37 °C in a humidified incubator containing 5% CO_2_.

For cell differentiation, the proliferation medium was changed, 24 h after seeding, with the differentiation medium containing 1:1 DMEM/F-12 (Ham), supplemented with 1% FBS, 1% modified Eagle’s medium nonessential amino acids, 0.5% P/S and 1 μM all-trans retinoic acid (RA) [6]. The differentiation medium was changed every two days. After 5 days of RA treatment, their morphology was changed and became neuron-like cells with interconnected neurites.

### 2.3. Macrophage Cells Culture

RAW 264.7, a murine macrophage cell line obtained from the American Type Culture Collection (ATCC) was cultured in DMEM-high glucose medium (Sigma-Aldrich; Merck KGaA, Darmstadt, Germany) supplemented with 10% fetal bovine serum (FBS; Sigma-Aldrich; Merck KGaA) and 1% penicillin/streptomycin, in atmosphere of 5% CO_2_ at 37 °C.

For the production of the LPS-stimulated medium, cells were growth until 80% confluence and incubated with 1 µg/mL LPS from Escherichia coli 0111: B4 (Sigma-Aldrich; Merck KGaA) for 24 h. Untreated cells were used as control. After the treatment, the culture medium was collected to carry out experiments with NSC-34 differentiated cells [24].

### 2.4. NSC-34 Treatment

After cell differentiation, the culture medium of NSC-34 cells was replaced with RA-free medium and treated with different concentrations of CBG (2.5 and 5 µM) and/or CBD (2.5 and 5 µM) alone and in 1:1 ratio combination and with vehicle (DMSO < 0.1%) for 24 h. After 24 h, the medium was replaced with a culture medium from LPS-stimulated Raw264.7 cells and the cannabinoid treatments restored. The cells were incubated for a further 24 h. As controls, NSC-34 cells were incubated with the medium of unstimulated RAW 264.7 macrophages.

### 2.5. Determination of Cell Viability by MTT Assay

An MTT assay was preliminarily performed in order to evaluate cell viability after treatment with various concentrations of CBG and CBD, both alone and in association for 48 h. Subsequently, we investigated the effect of exogenously applying the concentrations to the LPS-stimulated medium on the survival of NSC-34 differentiated cells and the effect of the pre-treatment with cannabinoids on counteracting the reduction in cell viability induced by LPS-stimulated medium exposure.

The NSC-34 cells were seeded into 96-well plates with 5 × 10^3^ cells per well and maintained at 37 °C in 5% CO_2_ in a proliferation medium. After 24 h, the culture medium was replaced with the differentiation medium and 1 µM of RA for 5 days. After differentiation, the cells were pre-treated with cannabinoids 24 h before incubating with the LPS-stimulated medium.

After the stimulus, 10 µL of methyl tetrazolium (MTT; 3-(4, 5-dimethylthiazol-2-yl)-2, 5-diphenyltetrazolium bromide) solution (5 mg/mL) was added to each well and cells were incubated for another 4 h. The formazan crystals were dissolved in 100 µL of the solubilization solution (0.1 N HCl in isopropanol) and the absorbance was measured at 490 nm.

Cells treated with vehicle (DMSO) and without LPS were included as controls. The experiments were performed in triplicates and repeated three independent times.

### 2.6. Protein Extraction and Western Blot Analysis

NSC-34 motor neurons were harvested with trypsin-EDTA, and proteins were extracted using NE-PER nuclear and cytoplasmatic extraction reagent (Thermo Scientific, Rockford, IL, USA) according to the manufacturer’s protocol. Cytoplasmic or nuclear protein extracts (30 µg) were separated using polyacrylamide gel electrophoresis (SDS-PAGE) and after transferred to an immun-blot PVDF membrane (Amersham Hybond, GE Healthcare Life Sciences, Milan, Italy). Subsequently, the membranes were incubated overnight at 4 °C with the following primary antibody:-anti-TNF-α (1:500; Cell Signaling Technology, Danvers, MA, USA);-anti-Nrf2 (1:100; Santa Cruz Biotechnology, Dallas, TX, USA);-anti-IL-10 (1:100; Santa Cruz Biotechnology);-anti-IL-37 (1:100; R&D System, Minneapolis, MN, USA);-anti-Nf-kB p65 (1:500; Cell Signaling Technology);-anti-Ikb alpha (1:500; Cell Signaling Technology);-anti-iNOS (1:50; Santa Cruz Biotechnology);-anti-Bax (1:500; Cell Signaling Technology).

Membranes were washed with phosphate saline buffer (PBS) and incubated with anti-rabbit or anti-rat or anti-goat or anti-mouse IgG peroxidase-conjugated secondary antibody (1:2000; Santa Cruz Biotechnology, Inc.) for 1 h at room temperature. The membranes were also incubated with the antibody for glyceraldehyde 3-phosphate dehydrogenase (GAPDH) HRP conjugated (1:1000; Cell Signaling Technology) to normalize protein values, removing loading bias. The relative expression of protein bands was analyzed through an enhanced chemiluminescence system (Luminata Western HRP Substrates; Millipore, Burlington, MA, USA). Protein bands were acquired using a ChemiDoc™ MP System (Bio-Rad Laboratories, Inc., Hercules, CA, USA) and quantified through ImageJ software (National Institutes of Health, Bethesda, MD, USA). All blots are representative of three independent experiments.

### 2.7. Immunocytochemistry

Nsc-34 motoneurons were seeded and differentiated on coverslips (12 mm, Thermo Fisher Scientific, Waltham, MA, USA). At the end of treatment, cells were fixed with 4% paraformaldehyde for 30 min, washed in PBS, and incubated with 3% hydrogen peroxide to suppress endogenous peroxidase activity. After a wash, the coverslips were blocked with horse serum 0.1% Triton X-100 for 20 min and incubated overnight at 4 °C with the following primary antibodies:-anti-PPARγ (1:50; Santa Cruz Biotechnology);-anti-Bcl2 (1:250; Cell Signaling Technology).

Afterwards, cells were washed and incubated with biotinylated secondary antibody (1:200; Vector Laboratories, Inc., Burlingame, CA, USA) and streptavidin AB Complex-HRP (ABC-kit from Dako, Glostrup, Denmark).

The immunostaining was developed with the DAB peroxidase substrate kit (Vector Laboratories, DBA Italia S.r.l., Milan, Italy; brown color; positive staining) and counterstaining with nuclear fast red (Vector Laboratories, DBA Italia S.r.l.; pink background; negative staining).

The images were captured using light microscopy (LEICA DM 2000 combined with LEICA ICC50 HD camera), and the densitometric analysis was carried out using LEICA Application Suite software ver. 4.2.0 (LEICA, Wetzlar, Germany). Quantitative analysis for each antibody was performed on 6 coverslips (three independent experiments, each in duplicate) by covering approximately 90% of the total area.

### 2.8. Statistical Data Analysis

Statistical analysis was performed using the GraphPad Prism version 8.0 computer software program (GraphPad Software, La Jolla, CA, USA). The data were statistically analyzed by one-way ANOVA test and Tukey’s post hoc test for multiple comparisons. A *p*-value less than or equal to 0.05 was considered statistically significant. Results are reported as the mean ± SEM.

## 3. Results

### 3.1. Cell Viability in Differentiated NSC-34 Motor Neurons Exposed to Different CBG and CBD Concentrations

NSC-34 cells, differentiated with RA 1 μM and low serum conditions (1%) for 5 days, showed neurite outgrowth (Figure 1b) compared to undifferentiated cells (Figure 1a).

In differentiated NSC-34 cells, we evaluated by an MTT assay the possible toxicity on motoneurons of serial concentrations (2.5, 5, 10, 20, 40, and 80 µM) of CBG and CBD alone and in association with a 1:1 ratio. No significant variation on cell viability was observed at all doses tested (Figure 2). We also analyzed NSC-34 cells incubated with dimethyl sulfoxide (DMSO < 0.1%), and no cytotoxicity was observed (data not shown) in line with our previous study reporting a normal morphological evaluation and nothing was altered in the transcriptome analysis of cells treated with DMSO [25]. All the experiments were performed in triplicates and repeated three independent times.

### 3.2. Cytokines Modulation

Motoneurons exposed to the LPS-stimulated medium showed significantly higher levels of tumor necrosis factor (TNF)-α compared to unstimulated cells (*p* < 0.0001) (Figure 3a). CBG pre-treatment at all doses tested, both alone and in association with CBD, was able to significantly downregulate TNF-α expression (*p* < 0.0001). On the contrary, the CBD result was less effective at 2.5 µM (Figure 3a).

As shown in Figure 4b, NSC-34 cells exposed for 24 h to the LPS-stimulated medium expressed significantly lower levels of IL10, the anti-inflammatory cytokine, compared to unstimulated cells. Only the pre-treatment of cells with CBG 5 µM + CBD 5 µM leads to an increase in basal levels of IL 10 cytokine (Figure 3b). CBD, both alone and in the association with CBG, was able to upregulate in a statistically significant manner the IL-37 expression compared to the LPS group (Figure 3c). CBG administered alone was not effective in restoring the anti-inflammatory cytokines but was able to enhance the anti-inflammatory effect of CBD.

### 3.3. Effect of CBG and CBD Association on IκBα Degradation, and NF-κB Activation

Exposure of differentiated NSC-34 cells to the LPS-stimulated medium induced a significant degradation of cytoplasmic IκBα (*p* < 0.0001), as well as a significant NF-κB nuclear translocation (*p* < 0.0001) compared to unstimulated cells. Only the association CBG 5 µM + CBD 5 µM was able to inhibit NF-κB nuclear translocation correlated with high levels of cytoplasmic IκBα (*p* < 0.0001). Instead, the association of CBG 2.5 µM + CBD 2.5 µM was able to inhibit only NF-κB nuclear translocation (*p* < 0.0001), suggesting a dose-dependent effect (Figure 4).

### 3.4. Antioxidant Action

Western blot analysis displayed a significant inducible nitric oxide synthase (iNOS) increased expression in cells incubated with the LPS-stimulated medium compared to the control cells (Figure 5). CBG 2.5 µM slightly reduced iNOS expression compared to the LPS group, but the pre-treatment with the association of CBG 5 µM + CBD 5 µM was able to counteract the iNOS expression. Besides, CBD at 2.5 µM and 5 µM doses both alone and in association with CBG, resulted in Nrf2 nuclear translocation assessed by Western blot analysis (*p* < 0.0001) (Figure 6). CBG alone, instead, was unable to activate the Nrf2 nuclear translocation and in combination therapy, did not potentiate the antioxidant effect of CBD.

### 3.5. Modulation of Apoptosis

The LPS-stimulated medium induced neuroinflammation characterized by a loss of viability on differentiated motoneurons with apoptosis induction. Western blot showed a significant increase in Bax expression in the LPS group compared to the control group (Figure 7; *p* < 0.0001).

The pre-treatment with all doses of CBG alone and in association with CBD at a 2.5 µM dose reduced Bax expression levels (Figure 7). In addition, immunocytochemistry showed negative staining for Bcl-2 in the LPS group (Figure 8B) compared to the control group (Figure 8A). The pre-treatment with all doses of CBG induced the expression of Bcl-2 (Figure 8C,D). Instead, CBD given alone showed negative staining of Bcl-2 at the 2.5 µM dose (Figure 8E), being effective only at the 5 µM dose (Figure 8F). Both CBG–CBD combinations induced the expression of the Bcl-2 factor (Figure 8G,H) in a dose-dependent manner.

### 3.6. PPARγ Activation

We also investigated the expression of PPARγ, assessed by immunocytochemistry. The pretreatment with CBG, both alone and in combination with CBD at all doses tested, induced PPARγ expression in motoneurons exposed to the LPS-stimulated medium (Figure 9). Only the activity of CBD is dose-dependent when administered alone (*p* < 0.0001).

## 4. Discussion

Neuroinflammation is an important factor in the pathogenesis of ALS; hence, in recent years, much effort has been focused on finding compounds effective to counteract inflammation [26,27,28].

Many preclinical studies have investigated the neuroprotective effects of cannabinoids [17], but to date, no study with CBG-CBD combination therapy in ALS is available. Our study aimed to evaluate the effectiveness of CBG and CBD administered alone and in combination, on an in vitro model of ALS performed on an NSC-34 cell line differentiated with retinoic acid. However, like any in vitro model, it has limitations that also depend on the time and type of compound used for differentiation, making them more or less susceptible to different neurotoxins [29].

We evaluated by MTT assay the possible toxicity on motoneurons of different concentrations (2.5, 5, 10, 20, 40, and 80 µM) of CBG and CBD administered both alone and in an association. At all concentrations tested, no toxicity was observed. In previous studies, we described the anti-inflammatory and antioxidant in vitro efficacy of CBG at a 7.5 µM dose and CBD at a 5 µM dose administered alone. Based on these results, we decided to evaluate the possible effects of the two lowest doses of CBG and CBD given alone and in combination, in order to choose a dose to daily administered in a mouse model of ALS.

The LPS-stimulated medium also induced neuroinflammation on motoneurons upregulating TNF-α and downregulating IL-10, similar to what was reported in our previous study [24].

TNF-α is a pleiotropic cytokine with homeostatic, immune, and inflammatory functions. At low levels, TNF-α exerts beneficial homeostatic functions, enhancing host defense mechanisms against intracellular pathogens. On the other hand, at high concentrations, TNF-α promotes inflammation and injury. TNF-α is a key mediator in the pathological mechanisms underlying many neurological disorders, including ALS, and autoimmune disorders such as rheumatoid arthritis, ankylosing spondylitis, and Crohn’s disease [30]. A recent study showed a significant increase in cerebrospinal fluid (CSF) TNF-α concentration in patients with ALS [31]. In ALS, plasma concentrations of TNF-α, TNF-Receptor 1, and TNF-Receptor 2 are found increased already at disease onset and remain over the normal range during the disease progression time [32]. However, a TNF-α gene knockout does not affect the life span or the extent of motoneuron loss in the superoxide dismutase 1 (SOD1) transgenic mice model of ALS, thus suggesting that TNF-α alone is not a key factor in motoneuron degeneration.

Notably, CBG pre-treatment, both alone and association with CBD at all doses tested, was able to reduce neuroinflammation, with the most effective treatment being the 5 µM dose of CBD administered alone, decreasing TNF-α levels; on the contrary, CBD was less effective at a 2.5 µM dose. According to Facchinetti F. et al., the effect of CBG at a 5 µM dose was able to counteract the release of TNF-α in LPS-stimulated rat microglial cells [33]. Petrosino et al. instead reported the ability of CBD to reduce TNF-α levels in HaCaT cells [34]. Moreover, in our study CBD, alone and in combination at all doses with CBG, increased the expression of IL-37 cytokines whereas CBG administered alone was not effective in restoring the anti-inflammatory cytokine. Only the pre-treatment of cells with CBG 5 µM + CBD 5 µM led to an increase in basal levels of IL 10 cytokine, as reported in a study conducted by our research group [35].

The high levels of TNF-α in LPS-stimulated motoneurons determine the activation of NF-kB, an inducible transcription factor activated by inflammation, which regulates the expression of several genes involved in immune and inflammatory responses [36]. In the inactivated state, NF-kB is complexed in the cytoplasm with the inhibitory protein inhibitor of kappaB alpha (IkBα). Extracellular stimuli, such as ROS and pro-inflammatory cytokines, degrade IκBα by site-specific phosphorylation by IκB kinase (IKK) complex. Phosphorylated NF-κB translocates into the nucleus and induces the expression of genes involved in the inflammatory response [37]. In our study, the data obtained from WB of nuclear extraction reported that only the combination of CBG 5 µM + CBD 5 µM blocked the nuclear translocation of NF-kB, and this result correlates with the previously reported decrease of TNF-α and the significant increase in the cytoplasmic IkBα. Instead, the association of CBG 2.5 µM + CBD 2.5 µM was able to inhibit only NF-κB nuclear translocation. We can state that the combination treatment prevented IκB-α phosphorylation and translocation of the nuclear NF-κB. The anti-inflammatory effects of CBD mediated by the NF-κB pathway have been previously proven in several in vitro models of neuroinflammation [38,39,40]. On the other hand, there is little evidence for CBG. In an in vitro model of oxidative stress, Giacoppo et al. described the effects of CBG at a 10 µM dose on reduced NF-κB activation [41]. Our results report how the treatment at the dose of 5 µM of CBD strongly inhibits the production of TNF-α, involved in the NFKB activation. The TNF-α inhibition is highly effective in preventing cytoplasmic activation and NF-kB nuclear translocation. The action of CBD is amplified with the co-association with CBG at the dose of 5 µM, inhibiting the activity of NF-kB; moreover, the association of CBD and CBG at the dose of 5 µM is strongly effective in stimulating the expression of anti-inflammatory cytokines IL-37 and IL-10. Therefore, the CBD treatment prevented the activation and translocation of NF-kB, thereby inhibiting the inflammatory response, with a TNF-α reduction and the enhancement of IL-10 and IL-37 cytokines.

The LPS-stimulated medium has also determined an increase in the oxidative stress, as demonstrated by the upregulation of the iNOS expression. NOS enzymes synthesize nitric oxide, which can lead to peroxynitrite formation, with consequent DNA damage [42]. It is known that LPS may lead to an increase in iNOS expression [43]. The pre-treatment with CBG and CBD administered in association at a 5 µM dose were able to counteract iNOS while CBG 2.5 µM showed a reduction of iNOS expression. Besides, CBD, given alone at a 2.5 µM dose and in combination with CBG at a 5 µM dose, displayed an antioxidant effect associated with an increase of Nrf2 nuclear translocation. CBG administered alone was not effective to increase Nrf2. Nrf2 regulates cellular redox status through endogenous antioxidant systems. Under oxidative stress conditions, Nrf2 translocates to the nucleus, binds the antioxidant response element (ARE), and induces the expression of several genes involved in the cellular antioxidant and anti-inflammatory defense [44]. The present results display that the CBD alone at a 2.5 µM dose has the highest effect in counteracting oxidative stress compared to CBD at a 5 µM dose and administered in the association.

Furthermore, in our in vitro model, neuroinflammation induced apoptosis with a loss of motor neurons and the upregulation of Bax protein. Western blot analysis showed a significant increase for Bax in the LPS group compared to the control group. These results are in line with a study reporting that LPS induced apoptosis in primary neurons [45]. The pre-treatment with CBG alone (at 2.5 and 5 µM doses) and in combination with CBD at a 2.5 µM dose showed a reduction of Bax expression and upregulating the expression of Bcl-2. Instead, CBD given alone at all doses did not regulate Bax expression and showed a negative staining of Bcl-2 at 2.5 µM in line with cell viability test. We observed a positive staining of Bcl-2 only at a 5 µM dose of CBD. Both CBG–CBD combinations induced the expression of the Bcl-2 factor in a dose-dependent manner.

Bcl-2 family members regulate mitochondrial integrity by the controlled release of factors involved in caspase-independent apoptosis. The Bcl-2/Bax unbalance leads to changes in the mitochondrial membrane potential and structure with the subsequent induction of apoptosis [46]. On the other, CBD at the 5 µM dose was able to modulate the expression of the anti-apoptotic protein Bcl-2, confirming the efficacy we previously described at the 5 µM dose [35]. Interestingly, we observed that the pre-treatment with CBG (2.5 and 5 µM) low doses is effective to reduce apoptosis, compared to a previous work where CBG was used at the dose of 7.5 µM [23]. These results indicate the anti-apoptotic effects in a single treatment of CBG at all doses. The completely new results, for us and in the literature, concerning the low-dose combination show that the association of CBD and CBG is able to inhibit the apoptotic process.

To evaluate the mechanism by which CBG and CBD exert the anti-inflammatory action, we investigated the possible involvement of PPARγ. PPARγ is an isoform of peroxisome proliferator-activated receptors implicated in the regulation of fatty acid storage, glucose metabolism, cell growth, and differentiation and anti-inflammatory responses [47]. It is known that the activation of PPARγ has anti-inflammatory effects that are beneficial in central nervous system diseases presenting inflammatory processes, including ALS [48]. Anti-inflammatory effects of PPARγ may be explained by inhibition of the expression of inflammatory markers including NF-κB. PPARγ can be activated by phytocannabinoids that increase its transcriptional activity, mediating its anti-inflammatory effects with a mechanism CB-receptors independent [20]. It has been shown that treatment with PPARγ agonists extends survival and decreases motor neuron loss in a mice model of ALS [49]. In our study, CBG, alone, and in association with CBD, increased the expression of PPARγ at all doses tested and also, the effect of CBD appeared to be dose-dependent. The highest expression levels of PPARγ were obtained in the combination of CBG and CBD at a dose of 5 µM. These data confirm the neuroprotective effects of CBG and CBD PPARγ-mediated, described in our previous studies [23,50].

## 5. Conclusions

In conclusion, in the present study, we confirmed the anti-inflammatory, antioxidant, and anti-apoptotic effects of CBG and CBD previously described. In particular, CBD administered alone was more effective at a 5 µM dose. CBG instead, given alone at 2.5 and 5 µM doses, was not effective but in co-administration with CBD, enhanced its anti-inflammatory properties. In particular, the combination treatment prevented the activation and translocation of NF-kB, thereby inhibiting the inflammatory response with a TNF-α reduction and the enhancement of anti-inflammatory cytokines IL-10 and IL-37. The benefits shared by CBD and CBG are enhanced when they are combined, making the overall compound more effective at the dose of 2.5 µM already. Moreover, the combinations were able to upregulate anti-apoptotic markers such as Bcl2.

Further evaluation of the effects of a CBG–CBD combination on in vivo preclinical models of ALS is therefore warranted. Our results provide preliminary support for the potential therapeutic application of CBG and CBD in motoneuron degenerative diseases.

## Figures and Tables

**Figure 1 medicina-55-00747-f001:**
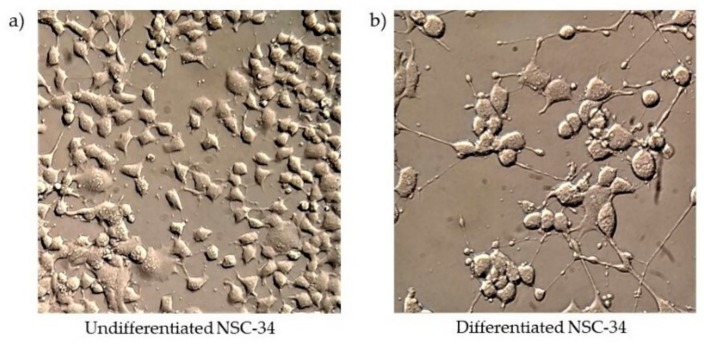
Morphological differentiation of motoneuron cell line NSC-34. (**a**) Undifferentiated cells (**b**) differentiated cells. The images were acquired with an objective 40×.

**Figure 2 medicina-55-00747-f002:**
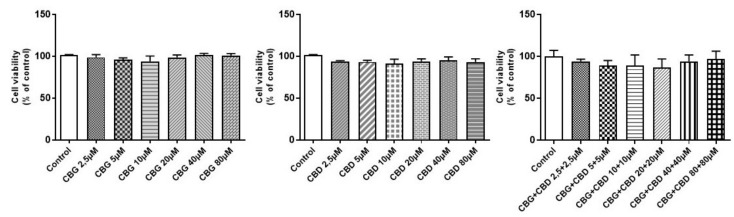
Cell viability in NSC-34 motor neurons differentiatedly exposed for 48 h to different cannabigerol (CBG) and cannabidiol (CBD) concentrations (2.5–80 µM). In all doses, no significant variation on cell viability was observed compared to the control group (*p* > 0.05). Data were expressed as mean (±SEM) of three independent experiments performed in triplicate.

**Figure 3 medicina-55-00747-f003:**
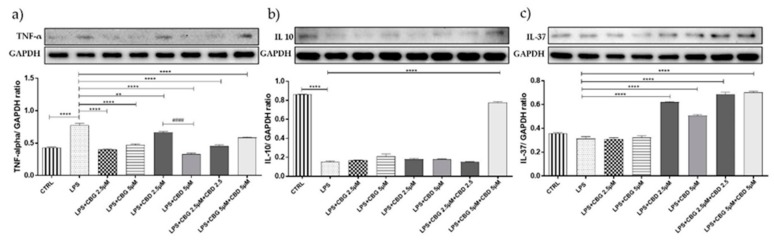
(**a**) Western blot analysis for TNF-alpha showed a downregulation of TNF-α expression in the pretreated groups ** *p* < 0.01, **** *p*  < 0.0001 vs. lipopolysaccharide (LPS); #### *p* < 0.0001 LPS + CBD 2.5 µM vs. LPS + CBD 5 µM. (**b**) Western blot analysis for IL-10 showed an increase in basal levels of the cytokine in the group pretreated with the CBG–CBD combination at 5 µM dose. **** *p* < 0.0001 vs. LPS. (**c**) Western blot analysis for IL-37 showed a statistically significant upregulation in groups pretreated with CBD alone and in association with CBG. **** *p* < 0.0001 vs. LPS. Blots are representative of three independent experiments. The densitometry analysis of bands representing means ± SEM of three independent experiments in which levels of each protein were normalized for glyceraldehyde 3-phosphate dehydrogenase (GAPDH).

**Figure 4 medicina-55-00747-f004:**
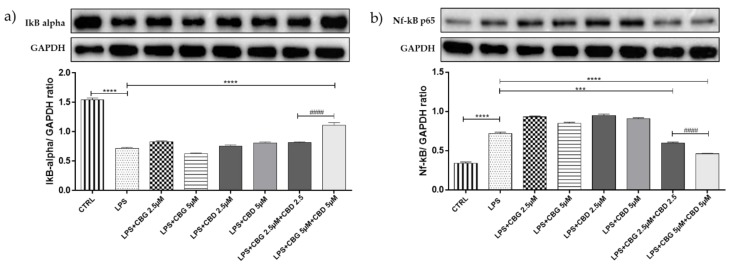
(**a**) Western blot analysis for cytoplasmic IkB alpha showed high levels of cytoplasmatic IkB alpha in the group pretreated with the CBG–CBD combination at 5 µM dose. (**b**) Western blot analysis for nuclear NF-kb p65 showed a significant NF-κB nuclear translocation in the groups pretreated with both CBG-CBD combinations. *** *p* < 0.001, **** *p* < 0.0001 vs. LPS; #### *p* < 0.0001 LPS + CBG 2.5 µM + CBD 2.5 µM vs. LPS + CBG 5 µM + CBD 5 µM. Blots are representative of three independent experiments. The densitometry analysis of bands representing means ± SEM of three independent experiments in which levels of each protein were normalized for GAPDH.

**Figure 5 medicina-55-00747-f005:**
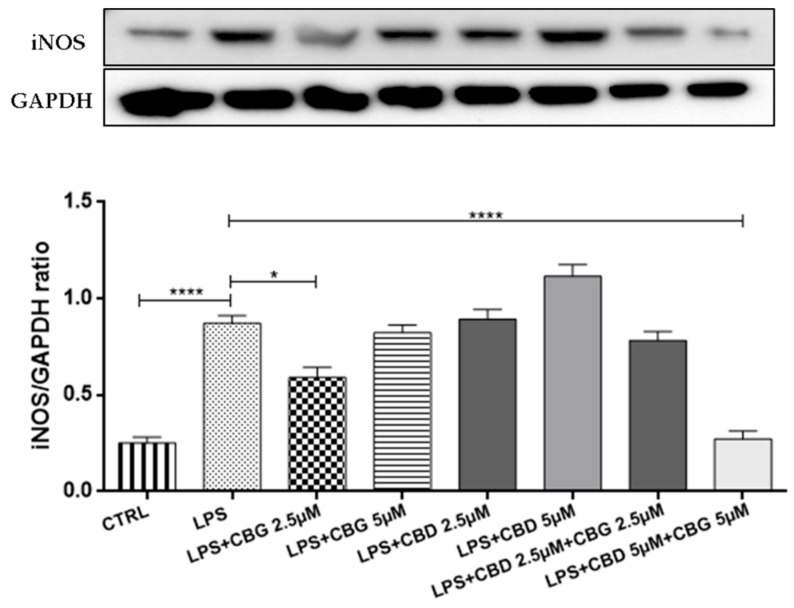
Western blot analysis for iNOS showed an reduction of iNOS expression in the CBG 2.5 µM group. The pre-treatment with the association of CBG–CBD at a 5 µM dose counteracted iNOS expression. * *p* < 0.05, **** *p* < 0.0001 vs. LPS. The blot is representative of three independent experiments. The densitometry analysis of bands representing means (±SEM) of three independent experiments in which levels of each protein were normalized for GAPDH.

**Figure 6 medicina-55-00747-f006:**
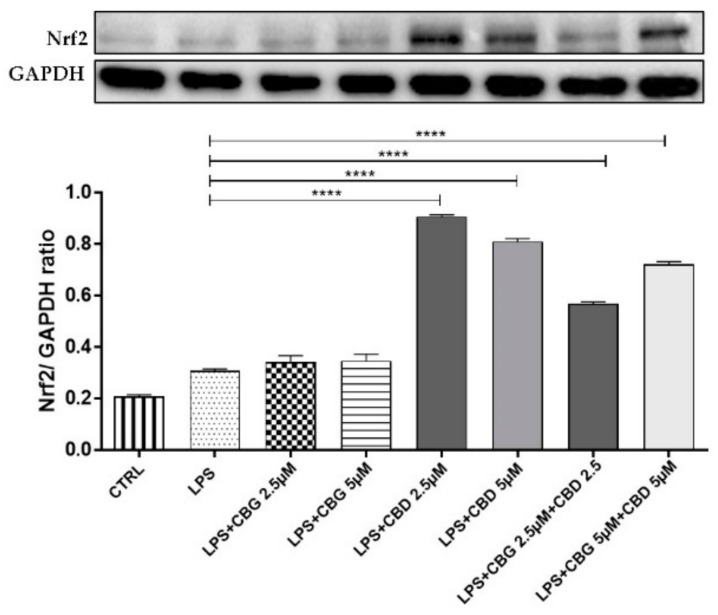
Western blot analysis for Nrf2 shows an increase of Nrf2 nuclear translocation in the groups treated with CBD alone at 2.5 and 5 µM doses and in combination with CBG at a 5 µM dose. **** *p* < 0.0001 vs. LPS. The blot is representative of three independent experiments. The densitometry analysis of bands representing means (±SEM) of three independent experiments.

**Figure 7 medicina-55-00747-f007:**
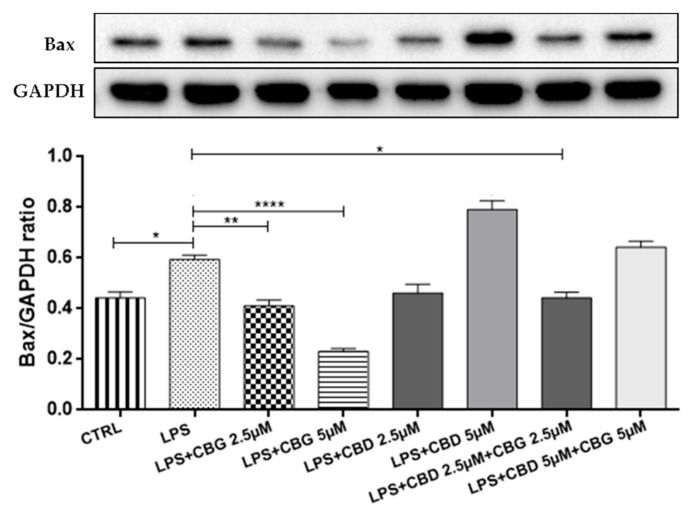
Western blot analysis for Bax. The pre-treatment with CBG alone at 2.5 and 5 µM doses and in combination with CBD at a 2.5 µM dose shows a reduction of Bax expression compared to the LPS group. * *p* < 0.05, ** *p* < 0.01, **** *p* < 0.0001 vs. LPS. The blot is representative of three independent experiments. The densitometry analysis of bands representing means (±SEM) of three independent experiments in which levels of each protein were normalized for GAPDH.

**Figure 8 medicina-55-00747-f008:**
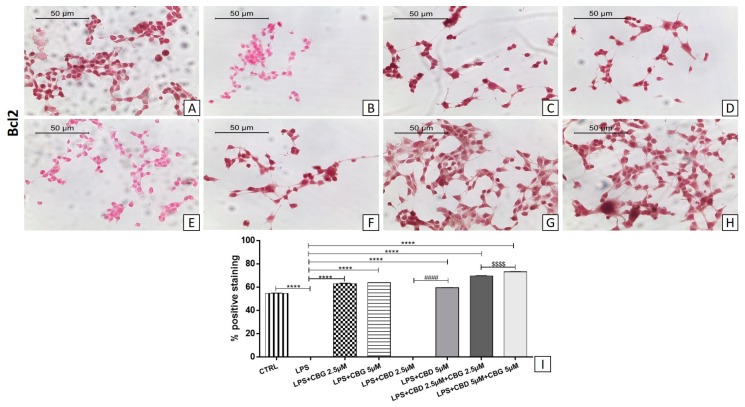
Immunocytochemical analysis for Bcl2 in untreated NSC-34 cells (CTRL) (**A**), cells incubated with the medium of LPS-stimulated macrophages (LPS) (**B**), LPS + CBG 2.5 µM (**C**), LPS + CBG 5 µM (**D**), LPS + CBD 2.5 µM (**E**), LPS + CBD 5 µM (**F**), LPS + CBG 2.5 µM + CBD 2.5 µM (**G**), LPS + CBG 5 µM + CBD 5 µM (**H**). Densitometric analysis for Bcl2 (**I**), **** *p* < 0.0001 vs. LPS; #### *p* < 0.0001 LPS + CBD 2.5 µM vs. LPS + CBD 5 µM; ^$$$$^
*p* < 0.0001 LPS + CBG 2.5 µM + CBD 2.5 µM vs. LPS + CBG 5 µM + CBD 5 µM. The data are representative of three independent experiments. Scale bar: 50 µm. The images were acquired with an objective 40×.

**Figure 9 medicina-55-00747-f009:**
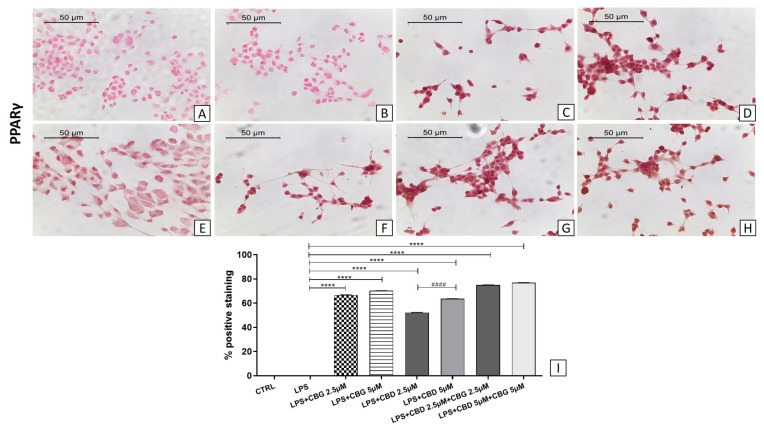
Immunocytochemical analysis for PPARγ in untreated NSC-34 cells (CTRL) (**A**), cells incubated with medium of LPS-stimulated macrophages (LPS) (**B**), LPS + CBG 2.5 µM (**C**), LPS + CBG 5 µM (**D**), LPS + CBD 2.5 µM (**E**), LPS + CBD 5 µM (**F**), LPS + CBG 2.5 µM + CBD 2.5 µM (**G**), LPS + CBG 5 µM + CBD 5 µM (**H**). Densitometric analysis for PPARγ (**I**), **** *p* < 0.0001 vs. LPS; ^####^
*p* < 0.0001 LPS + CBD 2.5 µM vs. LPS + CBD 5 µM. The data are representative of three independent experiments. Scale bar: 50 µm. The images were acquired with an objective 40×.

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
