# Peer review of "Could the Combination of Two Non-Psychotropic Cannabinoids Counteract Neuroinflammation? Effectiveness of Cannabidiol Associated with Cannabigerol"

_medicina, 2019, doi:10.3390/medicina55110747_

Round 1

Reviewer 1 Report

The current manuscript by Mammana et al. explore the role of CBG and CBD alone or in combination in in –vitro cell model of ALS. The authors reported that CBG+CBD both reduced inflammation and showed anti-apoptotic activities.Authors used NSC-34 cell lines to model ALS in vitro; however, this cell line does not necessarily mimic ALS (PMID: 27242431). Authors should discuss this caveat in the manuscript. I have some other concerns regarding this manuscript which are :

The manuscript should be thoroughly revised for English and grammar. There are many instances where syntax doesn’t look right. For instances in abstract-

“CBG 5µM, CBD 5µM and both the associations decreased iNOS expression and increased 25 the levels of the nuclear factor erythroid 2-related factor 2 (Nrf2).”” Our data showed anti-inflammatory, anti-oxidant and anti-apoptotic 28 effects peroxisome proliferator-activated receptor γ (PPARγ)-mediated”

Introduction reads in patches and should be rewritten in more coherent way. What were the purity% for CBG and CBD? The concentration of LPS used seems high. Usually 100ng/ml LPS is sufficient to cause inflammation. Any particular reason of using high dose? In cell viability by MTT, there is no difference observed for different doses of CBG or CBD od combination. Authors need to discuss this and should state the rationale behind the use of lowest two doses. What is the sample size in each experiment? Authors showed blots of secreted cytokines/interleukins in motor neuron. Usually, these cytokines are measured in culture supernatant because of their secretory nature. Did authors measure these in cell culture media at terminal point? It would be more meaningful in that case. The legends to figure should be self-explannatory to figures in brief. Please modify legends accordingly. Please provide the scale bar on Immunocytology images.

Author Response

Comments and Suggestions for Authors

The current manuscript by Mammana et al. explore the role of CBG and CBD alone or in combination in in –vitro cell model of ALS. The authors reported that CBG+CBD both reduced inflammation and showed anti-apoptotic activities.Authors used NSC-34 cell lines to model ALS in vitro; however, this cell line does not necessarily mimic ALS (PMID: 27242431). Authors should discuss this caveat in the manuscript. I have some other concerns regarding this manuscript which are:

R: Thank you for your suggestion.  According to literature we found 103 studies where NSC-34 cells are used. We repoterted in the discussion section the limitation of NSC-34 cell line,  from line 331 to line 333 and we reported the relative reference (Madji Hounoum B. et al 2016).

The manuscript should be thoroughly revised for English and grammar. There are many instances where syntax doesn’t look right. For instances in abstract-

“CBG 5µM, CBD 5µM and both the associations decreased iNOS expression and increased 25 the levels of the nuclear factor erythroid 2-related factor 2 (Nrf2).”” Our data showed anti-inflammatory, anti-oxidant and anti-apoptotic 28 effects peroxisome proliferator-activated receptor γ (PPARγ)-mediated”

R: Thank you for your suggestion. We have revised the manuscript for English and grammar and correct the sentences in line 26 and line 29.

Introduction reads in patches and should be rewritten in more coherent way.

R: Thank you for your suggestion. We have rewritten the introduction.

What were the purity% for CBG and CBD

R: Thank you for your suggestion. The purity for CBG and CBD is >99%. We have included this information in the main text in line 89.

The concentration of LPS used seems high. Usually 100ng/ml LPS is sufficient to cause inflammation. Any particular reason of using high dose?

R:  Thank you for your suggestion. We tested different doses of LPS and decided to perform the experiment with 1 µg/ml dose, which is also used in other independent studies. The medium of macrophage stimulated with LPS 100 ng was not able to induce toxicity in differentiated NSC-34. After differentiation with retinoic acid, the only dose capable of inducing mortality was 1 µg/ml dose according to the study of Wen W. et al 2006

In cell viability by MTT, there is no difference observed for different doses of CBG or CBD od combination. Authors need to discuss this and should state the rationale behind the use of lowest two doses.

R: Thank you for your suggestion. We decided to use the lowest dose that showed efficacy in order to be able to translate this study into an in vivo model of ALS and discuss this in “discussion section” from line 336 to 340

What is the sample size in each experiment?

R: Thank you for your question. The experiments were performed in triplicates and repeat for three independent times. We have included this information in the main text (material and methods section) and in all figure legends.

Authors showed blots of secreted cytokines/interleukins in motor neuron. Usually, these cytokines are measured in culture supernatant because of their secretory nature. Did authors measure these in cell culture media at terminal point? It would be more meaningful in that case.

R: Thanks for this comment.  We did not measure the cytokines in the cellular supernatant because it is the medium of LPS- stimulated macrophage and the dosage would not have been reliable.

The legends to figure should be self-explannatory to figures in brief. Please modify legends accordingly.

R: We have modified the figure legends according to your suggestion.

Please provide the scale bar on Immunocytology images.

R: As your request, we provide the scale bar on immunocytology images.

Reviewer 2 Report

In their work, Mammana et al. tried to unravel the anti-inflammatory mechanisms of 2 non-psychoactive cannabinoids, CBG and CBD, focusing on an in vitro ALS model.

This referee has some concerns about the designing of the work:

 “CBG and CBD exert their activity 53 at multiple molecular sites”. Please, describe and add references.

The authors evaluated cell viability with several concentrations of both compounds only without LPS. Higher concentrations than 5um would increase cell viability in LPS-sensitized neurons?

Pre-treatment is not clinically feasible.

Please, discuss the possible toxic/protective effects of DMSO.

How was Immunocytochemical analysis for iNOS in untreated NSC-34 cells performed? Did the authors only count positive cells? And what about the “intensity”? How did they discern between “intermediate” staining patterns? Would not be more appropriate to evaluate iNOS concentration/expression levels by means of WB and or PCR?

The same for BAX and Bcl2. This referee strongly recommends WB and/or PCR for these and previous markers. This is an important lack, as the paper relies on these experiments.

The only relative additive effect of combination therapy is the reduction of NF-kB nuclear factor activation, so, when considering future works, it seems that the combination therapy has showed no additive effect. Please, discuss.

TNF can develop both inflammatory and anti-inflammatory effects. Please, discuss.

The use of cell cultures in order to unravel the mechanisms by which cannabinoids act is limited. The use of in vivo models and KO animals is strongly suggested.

Author Response

Comments and Suggestions for Authors

In their work, Mammana et al. tried to unravel the anti-inflammatory mechanisms of 2 non-psychoactive cannabinoids, CBG and CBD, focusing on an in vitro ALS model.

This referee has some concerns about the designing of the work:

 “CBG and CBD exert their activity 53 at multiple molecular sites”. Please, describe and add references.

R: Thank you for your suggestion. We describe their activity at molecular sites from line 64 to line 77  and added the relative references (Morales, P. et al 2017; Bih, C.I. et al 2015; Jastrząb et al 2019; Fernández-Ruiz et al 2015; Navarro, G. et al 2018; Laprairie, R.B. et al 2015).

The authors evaluated cell viability with several concentrations of both compounds only without LPS. Higher concentrations than 5um would increase cell viability in LPS-sensitized neurons?

R: Thank you for your suggestion. We don’t have evaluated the viability of the other concentrations of cannabinoids on LPS- sensitized neurons because our goal was to find the lowest effective dose able to increase cell viability. 

Pre-treatment is not clinically feasible.

R: Thank you for your suggestion. CBG and CBD as two non-psychoactive compounds can be used as nutraceutical supplements

Please, discuss the possible toxic/protective effects of DMSO.

R: Thank you for your constructive criticism. In our studies, we tested cell viability following treatment with the different doses of DMSO corresponding to the related treatments. Being a concentration lower than 0.1%, as expected, was not toxic for the cells. We discussed in the main text from line 197 to 201 and we added the relative reference (Soundara R. et al 2017).

How was Immunocytochemical analysis for iNOS in untreated NSC-34 cells performed? Did the authors only count positive cells? And what about the “intensity”? How did they discern between “intermediate” staining patterns? Would not be more appropriate to evaluate iNOS concentration/expression levels by means of WB and or PCR?

R: The densitometric analysis for immunohistochemistry quantification was performed using the software “Leica application suite” provided by Leica. This software evaluates the 2 channels corresponding to the background and to the positive specific staining indicated by the brown color. For the densitometric analysis the software evaluates only the positive specific staining (brown color) that reaches the threshold value eliminating the background. As you requested we performed WB analysis for iNOS.

The same for BAX and Bcl2. This referee strongly recommends WB and/or PCR for these and previous markers. This is an important lack, as the paper relies on these experiments.

R: : Thank you for your constructive criticism. As you request we performed WB analysis for BAX. We are sorry but it was not possible to perform the WB analysis for Bcl2, unfortunately the necessary antibody did not arrive in time for the extension granted to us.

The only relative additive effect of combination therapy is the reduction of NF-kB nuclear factor activation, so, when considering future works, it seems that the combination therapy has showed no additive effect. Please, discuss.

R: We have discussed in the conclusion section from line 423 to 427.

TNF can develop both inflammatory and anti-inflammatory effects. Please, discuss.

R: Thank you for your suggestion. In our experimental model, TNF can develop only the inflammatory effect. We have discussed in the paragraph "discussion" from line 354 to 364 with relative references.

The use of cell cultures in order to unravel the mechanisms by which cannabinoids act is limited. The use of in vivo models and KO animals is strongly suggested.

R: Thank you for your suggestion. The in vivo models is the perspective of our work and will certainly be developed. These data provide only preliminary support useful for subsequent in vivo investigations on mouse models of ALS

Round 2

Reviewer 1 Report

The authors satisfactorily answered the raised concerns. I don't have further comments.

Author Response

Thank you for your consideration and positive response.

Reviewer 2 Report

1. Authors: "We don’t have evaluated the viability of the other concentrations of cannabinoids on LPS- sensitized neurons because our goal was to find the lowest effective dose able to increase cell viability."

It seems more reasonable to test higher concentrations until a “plateau” or no additive positive response is found. Then we can choose the lowest dose with highest efficacy. In other words, what if a higher dose could reveal higher cell viability? Would the authors use the lowest one or the one with best results?

2. Bax is not regulated with CBD nor with the combination therapy and this is not discussed. Moreover, these results seem controversial with the data obtained from cell viability, needing further discussion.

3. Is this statement correct? “The present results displayed that the CBG-CBD co-administration has a greater effect in counteracting oxidative stress than CBG and CBD given alone at 5 μM dose.” When checking figure 7, it seems that the highest effect was when using the 2.5uM CBD alone dose, then the 5uM CDB alone dose and then the combination dose.

4. What is “nutraceutical”? Is FDA approved?

5. The novelty of this paper relies on the comparison between the combination therapy and those individual regimens and this is not properly done in the discussion section. “Controversial” results must appear in discussion, such as the different protective results displayed by the therapies regarding the different markers evaluated (TNF, BAX, NFKB, etc.). This is part of the discussion.

6. Again, authors cannot state “the combination treatment prevented the activation and translocation of the NF-kB, thereby inhibiting the inflammatory response, with the TNF-α reduction and enhance of IL-10 and IL-37 cytokines.” Expression levels of TNF and IL-37 were reduced by other treatment options, so their decrease is not due to combination therapy. The combination effect of the therapy is limited, and this must not be hidden.

7. Please, revise English.

Author Response

Comments and Suggestions for Authors

Authors: "We don’t have evaluated the viability of the other concentrations of cannabinoids on LPS- sensitized neurons because our goal was to find the lowest effective dose able to increase cell viability."

It seems more reasonable to test higher concentrations until a “plateau” or no additive positive response is found. Then we can choose the lowest dose with highest efficacy. In other words, what if a higher dose could reveal higher cell viability? Would the authors use the lowest one or the one with best results?

  In our study, we have evaluated, by MTT assay, the possible toxicity on differentiated NSC-34 cells of several serial concentrations (2.5, 5, 10, 20, 40 and 80 µM) of CBG and CBD alone and in association with a 1:1 ratio. In previous work, different doses of CBG were tested. NSC-34 motor neurons were incubated 24h with the following CBG doses: 1, 2.5, 5, 7.5, 10, 12.5, 15 and 20 µM. In particular, for doses from 2.5 to 7.5 µM they observed about a 20% increase in cell viability compared to the control. On the bases of these results, they decided to perform the other experiments with CBG 7.5 µM (Gugliandolo A. et al 2018). Another study reported that hGMSCs treated with CBD at the concentration of 5 µM was not cytotoxic and did not cause cell death. Moreover displayed normal morphology. Otherwise, at 10 and 25 µM concentrations, CBD‐induced cell death in a dose-dependent manner (Rajan T.S. et al 2016). A recent study reported the viability of the SK-N-SH, NUB-6, IMR-32, and LAN-1 nbl cell lines after 24-48 hours of treatment with CBD. It reduced the viability of all cell lines in a dose and time-dependent manner, with CBD having a better effect on viability at the dose of 2.5 to 5 µM (Fisher T. et al 2016). On the bases of this study, we decide to evaluate 2.5 and 5 µM.

Bax is not regulated with CBD nor with the combination therapy and this is not discussed. Moreover, these results seem controversial with the data obtained from cell viability, needing further discussion.

 Thank you for your suggestion. We have discussed the effect of treatment with CBD in the discussion section. We apologize for the inconsistency but the image of vitality after treatment with LPS was found to be incorrect. We preferred to remove it.

Is this statement correct? “The present results displayed that the CBG-CBD co-administration has a greater effect in counteracting oxidative stress than CBG and CBD given alone at 5 μM dose.” When checking figure 7, it seems that the highest effect was when using the 2.5uM CBD alone dose, then the 5uM CDB alone dose and then the combination dose.

3  Thank you for your suggestion. We have modified the figure legends (modified in figure 6) at line 262 and the results in the discussion section from line 388 to 395 according to your suggestion.

What is “nutraceutical”? Is FDA approved?

  We apologize for the incorrect definition. our intent was to refer to a phito-compound. To date, the FDA has not validated any drug that includes the combination of CBD and CBG. At the moment there is only the approval of the Sativex by the FDA (CBD + delta 9 THC) for the treatment of spasticity in patients with multiple sclerosis

The novelty of this paper relies on the comparison between the combination therapy and those individual regimens and this is not properly done in the discussion section. “Controversial” results must appear in discussion, such as the different protective results displayed by the therapies regarding the different markers evaluated (TNF, BAX, NFKB, etc.). This is part of the discussion.

 Thank you for your suggestion. We have modified and reported the “controversial” results and the different protective doses for the different markers evaluated in the discussion section.

Again, authors cannot state “the combination treatment prevented the activation and translocation of the NF-kB, thereby inhibiting the inflammatory response, with the TNF-α reduction and enhance of IL-10 and IL-37 cytokines.” Expression levels of TNF and IL-37 were reduced by other treatment options, so their decrease is not due to combination therapy. The combination effect of the therapy is limited, and this must not be hidden.

 Thank you for your suggestion. We have modified the sentence and reported it in the discussion section from line 356 to 373.

Please, revise English.

 Thank you for your suggestion. We have revised English.
